# The effectiveness of a multi-domain electronic feedback report on the performance of quality indicators for chronic conditions: Protocol for a randomized controlled trial in general practice

**Levy Jäger** [ID]*, **Stefan Markun, Thomas Grischott** [ID]**, Oliver Senn** [ID]**, Thomas Rosemann** [ID]**, Jakob M. Burgstaller** [ID]

Institute of Primary Care, University Hospital Zurich, University of Zurich, Zurich, Switzerland

* levy.jaeger@usz.ch

**Data Availability Statement:** The datasets generated and analyzed during the current study

## Abstract

### Background

Chronic conditions are a significant public health concern due to their rising prevalence, association with high mortality, and substantial healthcare costs. General practitioners play a crucial role in managing these conditions, and quality indicators are essential tools for assessing the quality of care. Electronic feedback reports incorporating quality indicator performance have shown promise in improving care quality. However, most studies have focused on single conditions or link feedback to financial incentives, which may not sustain long-term practice changes. This study aims to evaluate the effectiveness of a multi-condition electronic feedback reports on quality indicator performance in Swiss general practice without financial incentives.

### Methods

This randomized controlled trial involves general practitioners enrolled in the FIRE project, a database of electronic medical records from Swiss primary care. Participants are randomized to receive either a plain feedback report or a comprehensive quality indicator -specific feedback report bi-monthly for 12 months. The plain feedback report contains descriptive summaries of practice activities, while the quality indicator-specific feedback report includes performance data on 14 quality indicators across cardiovascular, endocrine, pulmonary, and renal domains. The quality indicators were selected in multi-step process involving review of the literature and clinical guidelines, domain expert consultations, and a panel discussion with general practitioners. The primary study objective is to compare the effectiveness of the quality indicator-specific feedback report and of the plain feedback report with respect to the performance of the selected quality indicators.

cannot be shared publicly, as individuals or organizations with access to datasets that overlap with the FIRE database (such as health insurance claims data) may be able to identify patients in violation of legal restrictions on the identifiability of research subjects in Switzerland. Researchers who meet the criteria for access to confidential data can submit a request to the FIRE project team at the Institute of Primary Care of the University of Zurich at fire@usz.ch (see also our website https://www.fireproject.ch/en/kontakt).

**Funding:** This study is supported by a grant from the Federal Quality Commission (FQC) c/o Federal Office of Public Health (FOPH), Bern, Switzerland (https://www.bag.admin.ch/bag/de/home/das-bag/organisation/ausserparlamentarische-kommissionen/eidgenoessische-qualitaetskommission-eqk.html, contract number 142005597, awarded to TR). The funder had no role in study design, data collection and analysis, preparation of the manuscript, and decision to publish.

**Competing interests:** The authors have declared that no competing interests exist.

## Conclusion

The study addresses a critical gap by evaluating a multi-condition feedback report without financial incentives. Its findings can inform future health policies and strategies, in line with national and international initiatives that promote or even require the implementation of quality measurement activities in general practice.

## Trial registration

Trial registry: ISRCTN. Registration number: ISRCTN10637092, https://www.isrctn.com/ISRCTN10637092. Registered January 9, 2024.

## Introduction

Chronic conditions are an emerging public health priority. Their increasing prevalence, association with mortality and years lived with disability, and the resulting healthcare costs pose a challenge to healthcare systems worldwide [1]. Ensuring a high quality of services for patients with chronic conditions is therefore of utmost importance, especially for general practitioners (GPs), who are the first point of contact for many patients. The use of appropriate evidence-based preventive and therapeutic measures in general practice can significantly reduce progression, morbidity, and mortality [2].

In healthcare, quality indicators (QIs) are measurable elements of practice that can be used to assess quality of care in well-defined patient populations [3]. QIs can relate to a range of elements of care, such as structures, processes, or outcomes [4], and to a range of areas, such as prevention, acute, or chronic care. In general practice, QIs are a widely used tool to measure the quality of care for chronic conditions [5,6]. For example, QIs can address the delivery of preventive elements of care in the management of diabetes or cardiovascular diseases, such as vaccinations [7], glycemic control [8] or blood pressure control [9].

Audit and feedback interventions aimed at individual GPs have effectively incorporated QIs, and have shown the potential to considerably improve quality of care [10–12]. In particular, the use of electronic audit and feedback including information on QI performance has emerged as a promising tool, especially thanks to various advantages such as relatively low costs and scalability, enabled by the increasing availability of information infrastructures such as electronic medical records and primary care databases [12]. Several studies in general practice have examined the effect of electronic feedback reports (FBRs) that reflect information on QI performance at provider level. However, most studies have focused on single chronic conditions, such as hypertension or diabetes [12]. In other settings with more comprehensive sets of QIs, feedback is provided only annually, which may not be optimal for fostering continuous quality enhancement cycles. Moreover, most evaluations have rewarded QI achievement with financial incentives, which have been questioned for their potential to induce long-term practice change. This has led to calls for new incentive structures, potentially focusing on intrinsic motivations [13,14]. To date, there is a paucity of data on the effects of FBRs that report QI performance for multiple chronic conditions simultaneously without linking this information to financial incentives.

To address this gap, the trial presented in this protocol aims to determine the effectiveness of a multi-condition electronic FBR on QI performance in Swiss general practice without financial incentives.

## Materials and methods

### Setting and data source

Both the recruitment of GPs participating in this study and the data collection takes place within the Family medIcine Research using Electronic medical records (FIRE) project by the Institute of Primary Care of the University of Zurich [15]. The FIRE project includes a large database of electronic medical records provided by GPs across German-speaking Switzerland. It contains the following data: demographic characteristics (birth year and sex), consultation dates, reasons for encounters (coded according to the International Classification of Primary Care, ICPC-2 [16]), problems and diagnoses (mapped to the International Classification of Disease, ICD-10, system [17]), drug prescriptions (including Anatomic Therapeutic Chemical System [18] codes and Global Trade Identification Numbers [19]), blood pressure readings, biometric data (body height and weight), and laboratory test results. All data is fully anonymized on the patient level and is collected automatically on a daily basis from the participating practices using specifically developed interfaces.

A previous randomized controlled trial conducted among a subset of GPs participating in the FIRE project examined the effect of a pay-for-performance intervention targeting the quality of care delivered to patients with diabetes [20]. Although the intervention did not prove effective on the primary outcomes, the GPs in both study arms showed higher quality of care compared to non-participating GPs [21]. This difference was attributed to all participating GPs having received a bi-monthly electronic FBR containing detailed information on their performance with respect to quality of diabetes care during the study period. The FIRE project has therefore already shown potential as an appropriate setting to evaluate the effectiveness of electronic FBRs.

This protocol follows the Standard Protocol Items: Recommendations for Interventional Trials (SPIRIT) reporting guidelines for trial protocols (populated checklist in S1 Appendix) [22]. A SPIRIT schedule of enrollment, interventions, and assessments is provided in Fig 1.

### Definition of quality indicator performance

In this trial, we assess the performance of QIs addressing processes and outcomes in the following four specialty domains of medical practice: cardiovascular, endocrine, pulmonary, and renal. Each QI addresses care for one chronic condition from among asthma, atrial fibrillation or flutter, congestive heart failure, chronic kidney disease, chronic obstructive pulmonary disease, diabetes mellitus, and arterial hypertension. For each GP and QI, we first defined a QI-specific performance score ($PS_{QI}$), based on all patients seen by the GP during a given assessment period, as the percentage of patients who achieved the QI (numerator population) among all patients to whom the QI was applicable according to specified eligibility criteria (denominator population). We then defined a GP's overall performance score ($PS_O$) for the same assessment period as the average of all $PS_{QI}$s for that GP during the assessment period, weighted by the sizes of the respective denominator populations. Note that this definition of the $PS_O$ is equivalent to a score obtained as the ratio of the overall numerator population size, obtained as the sum of the numerator population sizes of all QIs, to the overall denominator population size, obtained as the sum of the denominator population sizes of all QIs. In addition, note that the same patient may be eligible for multiple QI denominator populations during a given assessment period. According to our definition, they are counted a corresponding multiple number of times for calculation of the $PS_O$.

To address performances in the different domains of chronic conditions, we also defined domain-specific performance scores ($PS_D$s). A GP's $PS_D$ for a given domain during a given

| | Enrollment | Allocation | STUDY PERIOD | | | | | | |
|---|---|---|---|---|---|---|---|---|---|
| | | | Post-allocation | | | | | | |
| TIMEPOINT | T-1 | T0 | T1 | T3 | T5 | T7 | T9 | T11 | T12 |
| **ENROLMENT:** | | | | | | | | | |
| **Eligibility screen** | X | | | | | | | | |
| **Opt-out** | X | | | | | | | | |
| **Allocation** | | X | | | | | | | |
| **INTERVENTIONS:** | | | | | | | | | |
| *QI-FBR delivery* | | | X | X | X | X | X | X | |
| *P-FBR delivery* | | | X | X | X | X | X | X | |
| **ASSESSMENTS:** | | | | | | | | | |
| *QI achievement, baseline period* | | X | | | | | | | |
| *QI achievement, follow-up period* | | | | | | | | | X |
| *Baseline characteristics* | | X | | | | | | | |

**Fig 1. Schedule of enrollment, interventions, and assessments.** T-1: The month preceding individual study start. T0: The day of individual study start (January 1, 2024, for GPs randomized in the first round, March 1, 2024, for the second round, and May 1, 2024, for the third round. T1, T3, T5, T7, T9, T11: Times of delivery of the electronic feedback reports (FBRs), chosen as appropriate dates that do not coincide with holidays or weekends during the corresponding months after T0. T12: The last day of data collection at the end of 12 months after T0, corresponding to individual study end (December 31, 2024, for GPs randomized in the first round, February 28, 2025, for the second round, and April 30, 2025, for the third round). Abbreviations: P-FBR, plain feedback report, QI, quality indicator; QI-FBR, quality indicator feedback report.

assessment period is calculated as the average of the $PS_{QI}$s of that GP belonging to that domain during that assessment period, weighted by the respective denominator population sizes.

## Quality indicator selection

The selection process for the QIs included in the study involved a comprehensive and iterative multi-step approach. Its aim was to identify a set of relevant and practicable QIs reflecting the above-mentioned specialty domains of medical practice.

To start, we thoroughly reviewed the existing literature, clinical guidelines, and best practices related to chronic conditions in the specified specialty domains of medical practice. The purpose was to identify potential QIs that had previously been validated or recommended in the literature and guidelines and that may be assessed in the FIRE database.

Following the literature review, we consulted a leading medical expert for each of the four specialty domains to gain insight and recommendations on the relevance and the potential impact of the QIs belonging to the respective domain.

In a next step, we engaged in discussions with a panel of six GPs practicing in different regions of German-speaking Switzerland who had extensive practical experience in providing care for the chronic conditions from all specified specialty domains. These GPs served as representatives from the study population and helped to gather perspectives and feedback on the perceived relevance, practicability and potential impact of the identified QIs. These discussions helped ensure applicability and acceptance of the selected QIs.

Based on the information gathered from the literature review, expert consultations, and panel discussion, the study team decided on the selection and definition of the QIs to be

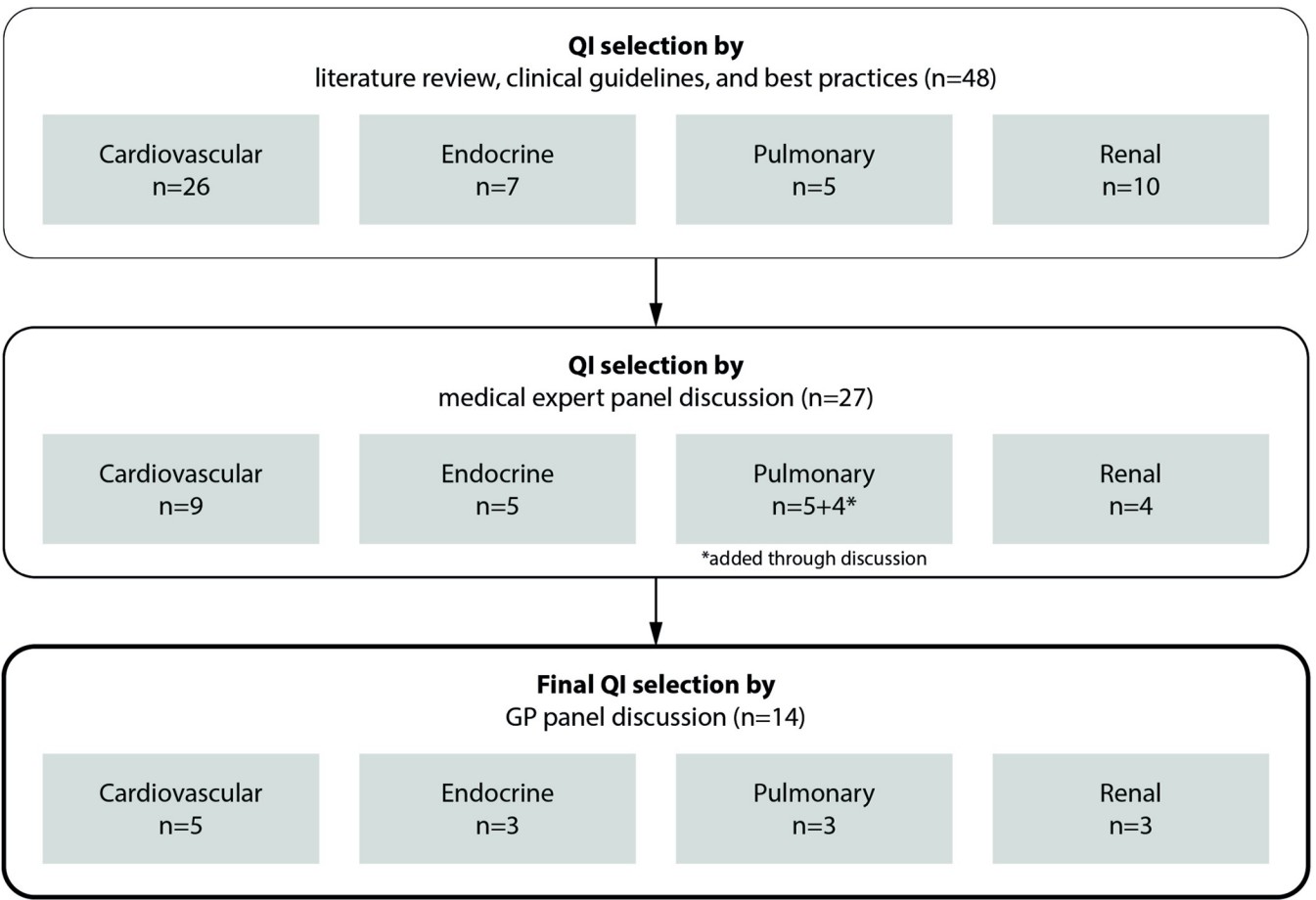

**Fig 2. Selection process of quality indicators.** Abbreviations: GP, general practitioner; QI, quality indicator.

included in the FBRs, aiming to balance scientific rigor, relevance, and acceptance. The QI selection process is summarized in Fig 2. The final definitions of the 14 selected QIs (5 in the cardiovascular, 3 in the endocrine, 3 in the pulmonary, 3 in the renal domain) are presented in Table 1. The operationalized criteria for the corresponding conditions mentioned in the definitions of the denominator populations and for QI achievement are presented in S2 Appendix.

## Design and interventions

In this study, we conduct a two-arm randomized controlled trial among GPs. GPs in both study arms receive personalized electronic FBRs every two months during a consecutive study period of 12 months, each containing information pertaining to an assessment period of 12 months prior to the respective time of FBR delivery. On the days of FBR delivery, participating GPs receive an email containing a password-protected link to their personalized electronic FBR as portable document format (PDF) documents, which can be viewed in an internet browser or downloaded.

The regular provision of personalized electronic FBRs with descriptive summaries of patient demography and events of care has already been established as a standard in the FIRE database. These plain FBRs (P-FBRs) do not contain any QI-specific information, but are intended as a tool for the GPs to monitor their practice activities over time, and thus as an

**Table 1. Definitions of the selected quality indicators.**

| No. | Domain | Type | Description |
|---|---|---|---|
| 1 | Cardiovascular | Outcome | Proportion of patients with hypertension aged <80 years whose latest BP measurement during the past 12 months was <140/90 mmHg (primary treatment target) or who were prescribed at least three antihypertensive drugs of different classes during the past 12 months. |
| 2 | Cardiovascular | Process | Proportion of patients with atrial fibrillation or atrial flutter and with risk factors for a thromboembolic event who received oral anticoagulation during the past 12 months. |
| 3 | Cardiovascular | Process | Proportion of CHD patients without other antithrombotic agents who received antiplatelet agents during the past 12 months. |
| 4 | Cardiovascular | Process | Proportion of CHD without diabetes and without CKD in stage G5[a] who received statins during the past 12 months. |
| 5 | Cardiovascular | Process | Proportion of CHD patients whose smoking status was documented during the past 12 months. |
| 6 | Endocrine | Process | Proportion of diabetes patients who received at least 2 HbA1c measurements at least 4 months apart during the past 12 months. |
| 7 | Endocrine | Process | Proportion of diabetes patients who received an influenza vaccination during the last vaccination period (October to December). |
| 8 | Endocrine | Outcome | Proportion of diabetes patients whose latest HbA1c during the past 12 months lied below an adapted treatment target. |
| 9 | Pulmonary | Process | Proportion of asthma patients with active controller therapy (long-acting beta2 agonist and/or long-acting muscarinic antagonist) who received inhaled corticosteroids during the past 12 months. |
| 10 | Pulmonary | Process | Proportion of COPD patients who received an influenza vaccination during the last vaccination period (October to December). |
| 11 | Pulmonary | Process | Proportion of asthma and/or COPD patients whose smoking status was documented during the past 12 months. |
| 12 | Renal | Outcome | Proportion of CKD patients not in stage G5 whose latest BP during the past 12 months was <140/90 mmHg. |
| 13 | Renal | Process | Proportion of CKD patients not in stage G5 who received at least one measurement of serum creatinine and BP during the past 14 months. |
| 14 | Renal | Process | Proportion of CKD patients not in stage G5 who received an angiotensin-converting enzyme inhibitor or an angiotensin receptor II blocker during the past 14 months. |

These quality indicators are included in the electronic feedback report of the intervention arm (QI-FBR). Abbreviations: BP, blood pressure; CHD, coronary heart disease; CKD, chronic kidney disease; COPD, chronic obstructive pulmonary disease; HbA1c, glycated hemoglobin. [a]Defined as an estimated glomerular filtration rate <15 ml/min/1.73m$^2$ [23].

incentive to participate in the FIRE project. GPs in the control arm of this study receive such P-FBRs containing information on the number of treated patients, demographic summaries, and descriptive summaries of the frequencies and results of selected laboratory tests, blood pressure readings, chronic conditions, and drug prescriptions occurring during the respective assessment period. Participants of the FIRE project did not receive any P-FBRs during the years 2022–2023, since usability of the FIRE database was limited due to updates and technical innovations during this time. Therefore, neither the GPs already enrolled in the FIRE project nor newly enrolled GPs had any recent exposure to electronic FBRs at study start.

GPs in the intervention arm receive comprehensive individualized FBRs (QI-FBRs) including information concerning the 14 QIs selected as described in the previous section. For the respective assessment period, each QI-FBR includes all 14 PS$_{QI}$s of the recipient, the changes of the PS$_{QI}$s with respect to the previous assessment period, and the distribution of the PS$_{QI}$s across all GPs in the FIRE database.

Both the P-FBR and the QI-FBR are two-page PDF documents presenting information with a mix of infographics and tables. Examples of a P-FBR and of a QI-FBR are shown in S3 and S4 Appendices, respectively.

## Participants and ethics

The participants were recruited among GPs formally enrolled in the FIRE project who have been contributing data to the FIRE database. For GPs enrolled in the FIRE project before December 31, 2023, we defined the study start at January 01, 2024. For GPs joining the FIRE project at later times, we defined individual study starts as the earliest date among March 01 or May 01, 2024, following enrolment in the FIRE project. All eligible GPs were notified of the

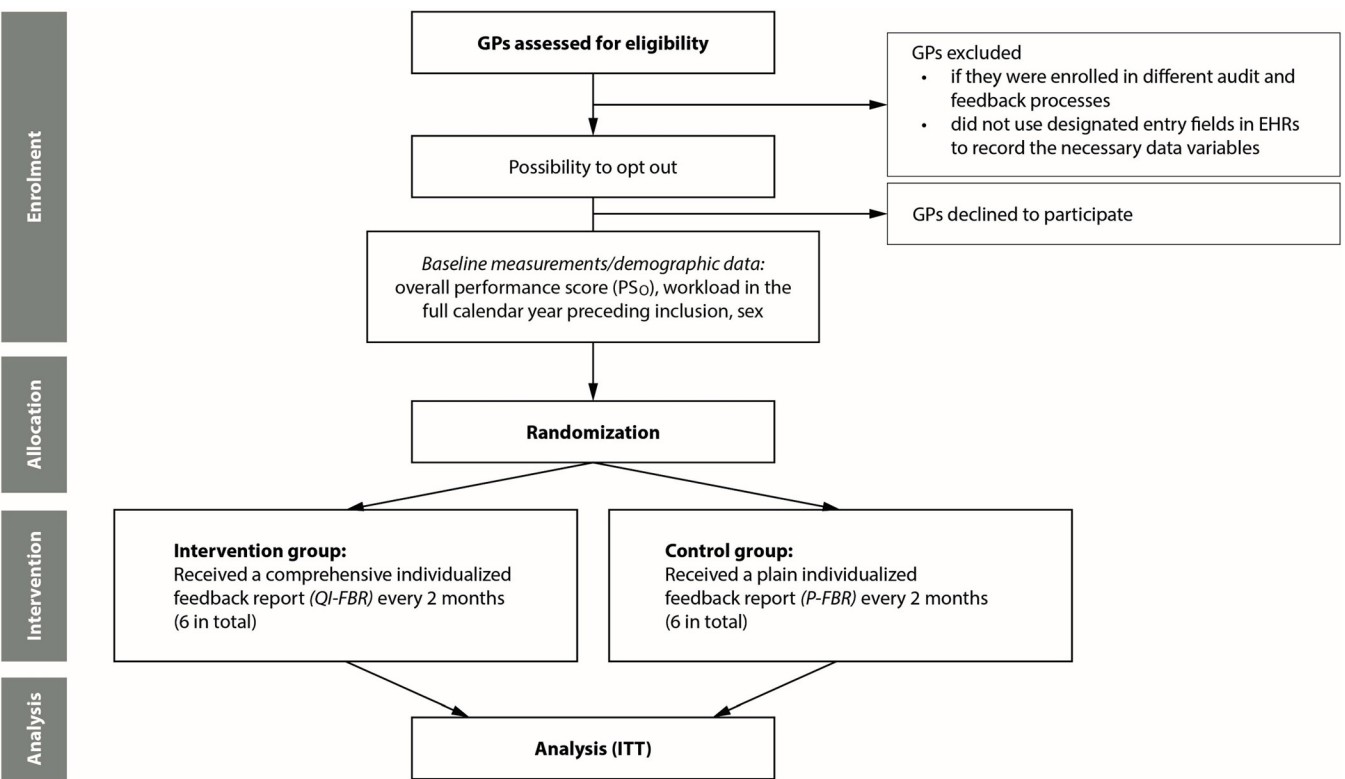

**Fig 3. Study flow chart.** Abbreviations: GP, general practitioner; ITT, intention to treat; QI-FBR, quality indicator feedback report; P-FBR, plain feedback report; $PS_O$, overall performance score.

study via email with an information sheet and were enrolled automatically, with the option to decline participation before randomization (opt-out approach).

To be eligible, GPs had to have worked in their respective practices for at least one year prior to their respective start of the study and be board certified as GPs (either in general internal medicine or in general medical practice). GPs were excluded if they did not use the designated entry field in their electronic medical records to record necessary data, such as diagnoses and laboratory values, or if they were already enrolled in different audit and feedback processes involving the FIRE project. The study flow chart is presented in Fig 3.

Due to the fully anonymized nature of patient data in the FIRE Project, patient consent is neither possible nor legally required, and the local Ethics Committee of the Canton of Zurich waived its approval in accordance with the Federal Act on Research Involving Human Beings (BASEC-Nr. Req-2023-01069). A copy of the study protocol version evaluated by the ethics committee is contained in S5 Appendix.

## Study period

For each GP, we defined their study period as the 12 months following their study start date. Depending on study start, the possible study end dates will thus be December 31, 2024, February 28, 2025, or April 30, 2025. During their study periods, GPs in both study arms receive links to their personalized electronic FBRs via e-mail on a bi-monthly basis. Each FBR uses the 12 full calendar months immediately preceding the day of delivery as the respective assessment period. Therefore, each GP receives six FBRs in total, the first in the first month immediately following the study start, and the last in the eleventh month of the study period. We choose the

times of delivery as suitable days of the corresponding months that do not coincide with national holidays or weekends. We further track the downloads of the electronic FBRs.

It is conceivable that members of physician networks may become aware of the FBR of the other study arm and then wish to receive that other FBR. In such a case, we comply with the request and provide the other FBR to that GP beginning with the next assessment period.

## Objectives

The primary study objective is to compare the effectiveness of the QI-FBR and of the P-FBR with respect to the overall QI performance. As secondary study objectives, we will compare the effectiveness of the QI-FBR and of the P-FBR with respect to QI performance in the different specialty domains considered.

## Outcomes

As the primary study outcome, we will use the change in the $PS_O$ between study start and study end. Secondary study outcomes will be the changes between the baseline and the follow-up period in domain-specific $PS_D$s.

## Randomization and blinding

The FBRs used as study interventions are personalized and sent individually to the participating GPs. We therefore anticipate a low potential for contamination and refrained from cluster randomization of practices. We allocated the participating GPs to the two study arms using a tool for covariate-constrained randomization to ensure balance with respect to $PS_0$ at baseline, sex, and workload in the full calendar year preceding inclusion (in terms of consultation count) [24]. A first round of randomization was performed in January 2024 before preparing the first FBRs for the GPs enrolled in the FIRE project who had not declined participation before December 31, 2023. A second and third round of randomization were carried out in March and May 2024, respectively, before preparing the corresponding FBRs for the GPs newly enrolled in the FIRE project who had not withdrawn their participation in the study until February 28, 2024, and April 30, 2024, respectively.

The participating GPs were informed about the study design, including random assignment to receive one of two different FBRs, but were not informed about the detailed differences between the two FBRs. However, we cannot completely rule out the possibility of partial unblinding between GPs within the same practice or quality circle for continuous quality improvement [25]. Members of the FIRE research team involved in preparation and dissemination of the electronic FBRs are not blinded to allocation. On the other hand, allocation is concealed from all remaining investigators, including the principal investigator and the study statistician.

## Statistical analysis

We will provide tabular summaries of the demographic and professional characteristics of included GPs at baseline, including distributions of QI performance.

We will analyze the primary study outcome using an intention-to-treat (ITT) framework, considering each GP according to the study arm to which they are randomized, regardless of whether they may have requested to receive the respective other FBR at some point during the study period. For each GP, we call the 12 months preceding their individual study start the baseline period and the 12 months from study start to study end the follow-up period,

respectively. We will use a binomial regression model in a dataset that includes, for each GP in the ITT population, one observation each for the baseline and follow-up periods. The total number of events and of successes will be given by the sizes of the corresponding denominator and numerator populations, respectively, for the corresponding GP, QI, and assessment period. We will use a mixed-effects model to account for each GP contributing two observations to the dataset (normally distributed GP-level random intercepts) and to account for unobserved effects on the practice level (normally distributed practice-level random intercepts) [26]. The model will include an indicator for the baseline versus the follow-up period, to assess the general effect of FBRs on QI performance, and an indicator for the QI-FBR arm versus the P-FBR arm during the follow-up period, to assess the differential effect of the QI-FBR in addition to the P-FBR on QI performance. In addition, we will adjust for the covariates used for balanced allocation (GP sex and workload). We will assess the primary study outcome by considering the statistical significance at the 0.05 level of the coefficient estimate associated with the indicator for the QI-FBR arm versus the P-FBR arm during the follow-up period.

GPs leaving the FIRE project, GPs whose practice eventually fails to export all relevant data to the FIRE database, and GPs who wish to opt out from the study during the study period after randomization will be considered as drop-outs.

## Power and sample size

Since we aimed to enroll as many GPs as possible among the GPs contributing to the FIRE project, we did not define a specific sample size to be achieved. Instead, we conducted a power analysis using model-based simulations to determine the statistical power resulting from different GP sample sizes and from different assumptions about the general effect of an FBR (any of the QI-FBR or P-FBR versus no FBR) on QI performance [27]. We based this power analysis on the intended final analysis described in the statistical analysis section and used estimates from a sample of 64 GPs enrolled in the FIRE project through 31 December 2023 who satisfied the eligibility criteria (details of the simulation in S6 Appendix). The overall QI performance across all GPs observed in this sample was 35.6%. We considered the following hypothetical levels of general FBR effects: OR of 1.0 (no effect), 1.3 (corresponding to the lower end of the FBR effects observed in the previously mentioned trial in the FIRE project involving diabetes care [21]), and 2.0 (corresponding to the upper end of the effects observed in this previous trial). For the differential effect of the QI-FBR versus the P-FBR, we assumed an absolute increase of 2 percentage points in the overall QI performance (corresponding to an OR of 1.09 in the observed sample), which is a small effect size compared to the 10 percentage point improvement often assumed in power analyses of previous studies involving performance feedback on similar QIs [28–31]. Relevant GP characteristics, including the adjustment covariates planned for the final analysis and the size of the denominator populations, were accounted for by bootstrapping from the observed sample of 64 GPs. With this constellation, assuming 1:1 allocation and with an alpha of 0.05, power was determined to be at least 88% with a total sample size of at least 50 GPs, regardless of assumptions about general FBR effects. Assuming a drop-out rate of 10%, which is slightly higher than that observed in the aforementioned previous trial within the FIRE project [20], a sample size of 56 GPs would ensure a statistical power of 88% to detect a 2 percentage point increase in overall QI performance across all GPs. We performed this power analysis using the statistical software R version 4.3.2 (R Foundation for Statistical Computing, Vienna, Austria) and used the lme4 package for mixed-effects modelling [32,33].

## Secondary analyses

We will conduct secondary analyses to examine the effect of the QI-FBR by chronic condition domain and by individual QI. In addition, we will examine the variation of the effect of the QI-FBR with different download frequencies. All secondary analyses will involve the regression model described in the statistical analysis section extended by appropriate interaction terms. We will address the global hypotheses of whether the QI-FBR is effective on performance in any of the chronic condition domains or in any of the QIs by considering the *p*-values associated with the inclusion of the respective interaction terms.

## Sensitivity analyses

When calculating the $PS_O$, we will assess the performance of the participating GPs by aggregating the single $PS_{QI}$s using denominator-based weights, a commonly used procedure for composite scoring also called overall percentage method in the literature [34]. However, some QIs may be more difficult to achieve, and if the corresponding patients are overrepresented in the populations treated by certain GPs, their scores may be unfairly lower compared to others. To address this concern, we will conduct a sensitivity analysis for the primary objective using an unweighted version u-$PS_O$ of the $PS_O$, defined, for a given GP and assessment period, as the unweighted arithmetic mean of the 14 $PS_{QI}$s of that GP during that assessment period.

In addition to the main ITT analysis, we will conduct a per-protocol sensitivity analysis involving the assessment of the primary outcome only considering the GPs who received the FBR as intended by randomization throughout the study period.

## Missing data

In the event that more than 5% of the randomized GPs will drop out during the study period, we will impute missing measurements for the follow-up period using multilevel chained equations-based multiple imputation by predictive mean matching with five datasets and 20 iterations using the R package micemd [35]. We will then present the results of the study based on both datasets without and with the imputed follow-up measurements.

## Trial status and timeline

The study started in January 2024 and is ongoing. The third and final randomization round took place in May 2024. Data collection will end on December 31, 2024. This manuscript describes version 3.4 of the protocol. This study protocol corresponds to the study protocol version 3.4.0 of 24 July 2024.

## Data management and safety

Information on QI performance is integrated into the FIRE database, which is only accessible to members of the FIRE core team. However, the database does not contain information on group allocation in the trial, and the process of generating and disseminating FBRs is decoupled from the database itself. The statistician performing the statistical analysis of this study will be blinded to group allocation of the participating GPs.

Data are exported daily and automatically from the participating practices to the FIRE database. Data in the FIRE database is securely stored and handled in accordance with local data protection regulations.

### Trial sponsor

This is an investigator-Initiated clinical trial conducted by the Institute of Primary Care of the University of Zurich. The sponsor is responsible for designing the study, collecting, managing, analyzing, and interpreting the data, writing the report, and deciding on the submission of the report for publication.

## Discussion

This protocol describes the rationale, methodology, and design of a randomized controlled trial to determine the effectiveness of a multi-condition FBR on QI performance in Swiss general practice.

QIs can objectively measure and compare various dimensions of quality of health care and have long been utilized for quality improvement in general practice [5]. Most notably, the development and implementation of QIs in healthcare settings were successfully demonstrated by the Quality and Outcomes Framework (QOF) in England, Wales, and Northern Ireland. Specifically, participation in the QOF is voluntary and GPs are incentivized by financial rewards granted if specific QI goals are met [6]. Despite the fact that approximately 95% of practices participate in the QOF and generate between 10% and 15% of their income with the QOF, the effectiveness of the financial incentives provided by the QOF to improve clinical outcomes is still a matter of debate [14]. In contrast to the QOF model, in our study, we hypothesize that a focused multi-condition FBR which is provided on a bi-monthly schedule can improve quality of care without additional financial incentives. Novel approaches of this kind are worth exploring, especially with evidence not supporting the effectiveness of financial incentives in Swiss primary care [20].

This project fits into a wider international context where there is an increasing focus on quality in primary care. For example, several initiatives across Europe aim to improve quality in primary care. The European Association of Quality and Safety in General Practice/Family Medicine (EQuiP) is such an initiative that focuses on improving quality and safety in primary care across Europe [36]. EQuiP aims to raise standards in primary care by developing and disseminating tools, guidelines, and training programs. In Germany, the Strengthening Primary Care Act (GKV-Versorgungsstärkungsgesetz) is a governmental program that aims to enhance outpatient care by increasing support for general practitioners, promoting care coordination, and providing financial incentives for physicians in underserved areas [37]. Similarly, in Switzerland, the National Strategy for the Prevention of Non-communicable Diseases aims to reduce the burden of non-communicable diseases through coordinated and comprehensive public health measures [38]. Unfortunately, several factors can act as barriers towards implementation of initiatives in general practice, including difficulties integrating interventions in the workflow, the political and policy environment, or economic and financial constraints [39,40]. In this regard, our study assesses a specific and feasible feedback process that builds on existing infrastructure and that does not disrupt the GPs' daily practice with additional documentation burden. These aspects would drastically facilitate the implementation of the QI-FBR intervention on a large scale.

Our study intervention involves electronic audit and feedback and uses the concept of benchmarking, which allows participating GPs to gain insight into how they perform relative to their peers. This facilitates self-assessment and identification of areas for improvement, which may in turn spur intrinsic motivation [41,42]. In addition, the use of electronic medical records brings the advantage of automated data collection and generation of FBRs, which ensures accurate and timely feedback. The QI performance data can also be used beyond the

scope of the FBRs to identify gaps in the quality of care provided to patients on level of the Swiss general practice setting.

Our study is also important in the context of a recent amendment to the Swiss Health Insurance Act (Article 58a) of 2021 [43], which obliges physicians to implement quality measurement and quality improvement activities. Such activities arguably add to the administrative burden already present in general practice [40]. The provision of electronic feedback, as implemented in the QI-FBRs, would allow GPs to comply with the new requirements with minimal disruption to their daily practice. Previous initiatives to enhance the quality of general practice in Switzerland have been varied, but have never achieved national coverage[40]. In particular, previous attempts to implement monitoring systems in Swiss general practice have been hampered by problems of data availability. [44] With its scalability, the FIRE project has a great potential to support quality development in Swiss general practice, for which the development of high-quality electronic FBRs constitutes an important step.

### Limitations

This study has potential limitations. Measuring the quality of care using data from electronic records has certain disadvantages, such as heterogeneous documentation habits of GPs and issues related to data quality [45,46]. In addition, it can be argued that the assessment of QI performance based on electronic records captures only limited dimensions of quality, as they typically do not contain patient-relevant outcomes or experiences [47]. In addition, patient involvement would be an important future step in the further development of the QIs used in this study.

While the FIRE project shows a considerable coverage of the German-speaking part of Switzerland [48], the other language regions of the country are not yet represented in the database. It is possible that this restriction limits the applicability of the results of this study at the national level.

A potential limitation of the study design is the possibility of partial unblinding between GPs within the same practice, as randomization occurred at the level of individual GPs and not as cluster randomization of practices. However, we do not expect any major spill-over effects as we provide individualized password-protected links to the electronic FBRs. On the other hand, cluster randomization would have presented additional challenges in randomization, analysis, and issues of reduced statistical power [49].

### Dissemination plans

We will disseminate the trial results in scientific peer-reviewed journals and at national conferences, regardless of the observed effect sizes and directions. We will report and discuss the results of this study with the participating GPs and the participants of the FIRE project. An additional report will be handed to the Federal Quality Commission (FQC) of the Federal Office of Public Health.

### Conclusions

This study addresses a critical gap in the literature on effective strategies for quality improvement in general practice. Its findings have the potential to significantly improve our understanding of the role of electronic FBRs in an environment where multiple conditions and multiple domains of medical care are involved in daily practice. The outcomes of this study can inform future health policies and can help implement quality improvement strategies in general practice settings.

## Supporting information

**S1 Appendix. Compiled SPIRIT reporting checklist.**
(PDF)

**S2 Appendix. Operationalized criteria for the identification of chronic conditions and for the quality indicator definitions in the FIRE database.**
(PDF)

**S3 Appendix. Example of a plain feedback report for the control arm (P-FBR).**
(PDF)

**S4 Appendix. Example of a quality indicator feedback report for the intervention arm (QI-FBR).**
(PDF)

**S5 Appendix. Study protocol evaluated by the ethics committee.**
(PDF)

**S6 Appendix. Details of the power analysis.**
(PDF)

## Acknowledgments

The authors thank Fabio Valeri and Raffael Golomingi of the Institute of Primary Care of the University of Zurich for maintaining the FIRE database. The authors also thank the GPs in the FIRE study group, especially the participants in the trial described in this protocol. Further acknowledgement goes to the experts who helped selecting suitable QIs from the diverse specialty domains: Prof. Dr. Felix Beuschlein (Department of Endocrinology, Diabetology and Clinical Nutrition, University Hospital Zurich), Dr. Harald Seeger (Department of Nephrology and Dialysis, Kantonsspital Baden), Prof. Dr. Dr. Jörg Leuppi (Department of Internal Medicine, Kantonsspital Baselland), and Dr. Dr. Roman Brenner (Department of Cardiology, Kantonsspital St. Gallen). We also thank the GPs who participated in the QI discussion panel for their valuable input: Dr. Roland Fischer (Seengen, AG), Dr. Jenny Studer (Eggersriet, SG), Prof. Dr. Dr. Sven Streit (Konolfingen, BE), Dr. Michael Fluri (Langendorf, SO), Dr. Eveline Breidenstein (Obfelden, ZH), Dr. Dr. Adrian Rohrbasser (Wil, SG).

## Author Contributions

**Conceptualization:** Levy Jäger, Thomas Rosemann, Jakob M. Burgstaller.

**Data curation:** Levy Jäger, Thomas Grischott, Jakob M. Burgstaller.

**Formal analysis:** Levy Jäger.

**Funding acquisition:** Thomas Rosemann, Jakob M. Burgstaller.

**Investigation:** Levy Jäger, Stefan Markun, Jakob M. Burgstaller.

**Methodology:** Levy Jäger, Thomas Grischott, Jakob M. Burgstaller.

**Project administration:** Thomas Rosemann, Jakob M. Burgstaller.

**Resources:** Thomas Rosemann.

**Software:** Levy Jäger.

**Supervision:** Oliver Senn, Thomas Rosemann.

**Validation:** Levy Jäger, Jakob M. Burgstaller.

**Visualization:** Jakob M. Burgstaller.

**Writing – original draft:** Levy Jäger, Jakob M. Burgstaller.

**Writing – review & editing:** Levy Jäger, Stefan Markun, Thomas Grischott, Oliver Senn, Thomas Rosemann, Jakob M. Burgstaller.

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
