## [Decision Letter · Decision Letter 0]

29 Oct 2024

PONE-D-24-36573The Effectiveness of a multi-domain electronic feedback report on the performance of quality indicators for chronic conditions: protocol for a randomized controlled trial in general practicePLOS ONE

Dear Dr. Jäger,

Thank you for submitting your manuscript to PLOS ONE. After careful consideration, we feel that it has merit but does not fully meet PLOS ONE’s publication criteria as it currently stands. Therefore, we invite you to submit a revised version of the manuscript that addresses the points raised during the review process. Please consider the concerns raised about your proposed sample size and analysis as per the comments below. 

We look forward to receiving your revised manuscript.

Kind regards,

Kathleen Finlayson

Academic Editor

PLOS ONE

Journal Requirements: When submitting your revision, we need you to address these additional requirements. 1. Please ensure that your manuscript meets PLOS ONE's style requirements, including those for file naming. The PLOS ONE style templates can be found at https://journals.plos.org/plosone/s/file?id=wjVg/PLOSOne_formatting_sample_main_body.pdf and https://journals.plos.org/plosone/s/file?id=ba62/PLOSOne_formatting_sample_title_authors_affiliations.pdf 2. In the online submission form, you indicated that your data is available only on request from a third party. Please note that your Data Availability Statement is currently missing contact details for the third party, such as an email address or a link to where data requests can be made. Please update your statement with the missing information.  3. Please include your full ethics statement in the ‘Methods’ section of your manuscript file. In your statement, please include the full name of the IRB or ethics committee who approved or waived your study, as well as whether or not you obtained informed written or verbal consent. If consent was waived for your study, please include this information in your statement as well. 4. Please review your reference list to ensure that it is complete and correct. If you have cited papers that have been retracted, please include the rationale for doing so in the manuscript text, or remove these references and replace them with relevant current references. Any changes to the reference list should be mentioned in the rebuttal letter that accompanies your revised manuscript. If you need to cite a retracted article, indicate the article’s retracted status in the References list and also include a citation and full reference for the retraction notice.

Reviewers' comments:

Reviewer's Responses to Questions

**Comments to the Author**

1. Does the manuscript provide a valid rationale for the proposed study, with clearly identified and justified research questions?

Reviewer #1: Yes

Reviewer #2: Yes

2. Is the protocol technically sound and planned in a manner that will lead to a meaningful outcome and allow testing the stated hypotheses?

Reviewer #1: Yes

Reviewer #2: Yes

3. Is the methodology feasible and described in sufficient detail to allow the work to be replicable?

Reviewer #1: Yes

Reviewer #2: Yes

4. Have the authors described where all data underlying the findings will be made available when the study is complete?

Reviewer #1: Yes

Reviewer #2: Yes

5. Is the manuscript presented in an intelligible fashion and written in standard English?

Reviewer #1: Yes

Reviewer #2: Yes

6. Review Comments to the Author

You may also provide optional suggestions and comments to authors that they might find helpful in planning their study.

Reviewer #1: The authors have submitted a protocol describing the implementation and evaluation of an audit & feedback RCT. They plan to estimate the impact of providing GPs specific quality improvement performance reports versus controls receiving a generic feedback report without quality indicators.

The protocol is extremely well-written and encompasses all essential components for evaluation. The statistical methods are of a high standard and demonstrate careful planning. I have no significant critiques regarding any section of this protocol.

Reviewer #2: I have the following concerns:

Sample size calculation:

How many patients were assumed per GP? 2% difference is small relative to 50 GPs.

2% difference needs to be clinically justified.

What assumptions were used for random factors?

Analysis:

What mixed model will you use? GEE or generalized linear mixed model?

Is there any multiple testing correction for secondary endpoints?

Missing:

Can you increase the sample size to allow for dropouts instead of imputation? (e.g., if expecting 10% dropout, recruit 55 GPs instead of 50)

7. PLOS authors have the option to publish the peer review history of their article (what does this mean?). If published, this will include your full peer review and any attached files.

Reviewer #1: No

Reviewer #2: No

---

## [Author Response · Author response to Decision Letter 0]

1 Nov 2024

Response to Journal Requirements

2. In the online submission form, you indicated that your data is available only on request from a third party. Please note that your Data Availability Statement is currently missing contact details for the third party, such as an email address or a link to where data requests can be made. Please update your statement with the missing information. 

Authors’ response: We have updated our Data Availability Statement as requested to include an email address and a link to the FIRE project website:

Data Availability Statement: Researchers who meet the criteria for access to confidential data can submit a request to the FIRE project team at the Institute of Primary Care of the University of Zur-ich at fire@usz.ch (see also our website https://www.fireproject.ch/en/kontakt).

Authors’ response: We have updated the ‘Participants and ethics’ section of the manuscript to in-clude the full name of the ethics committee that waived consent for our study:

Page 10, Lines 212-215: Due to the fully anonymized nature of patient data in the FIRE Project, patient consent is neither possible nor legally required, and the local Ethics Committee of the Can-ton of Zurich waived its approval in accordance with the Federal Act on Research Involving Hu-man Beings (BASEC-Nr. Req-2023-01069).

4. Please review your reference list to ensure that it is complete and correct. If you have cited papers that have been retracted, please include the rationale for doing so in the manuscript text, or remove these references and replace them with relevant current references. Any changes to the reference list should be mentioned in the rebuttal letter that accompanies your revised manu-script. If you need to cite a retracted article, indicate the article’s retracted status in the Refer-ences list and also include a citation and full reference for the retraction notice.

Authors’ response: Our only changes to the reference list address reviewer comments and are de-scribed in our responses below.

Response to Reviewer 1

Reviewer’s general comments: The authors have submitted a protocol describing the imple-mentation and evaluation of an audit & feedback RCT. They plan to estimate the impact of providing GPs specific quality improvement performance reports versus controls receiving a generic feedback report without quality indicators.

The protocol is extremely well-written and encompasses all essential components for evaluation. The statistical methods are of a high standard and demonstrate careful planning. I have no signif-icant critiques regarding any section of this protocol.

Authors’ reply: We thank the reviewer for their very encouraging compliments. We are pleased to hear that our efforts to ensure sound methodology are visible.

Response to Reviewer 2

Reviewer’s general comments: I have the following concerns:

Authors’ general reply: We thank the reviewer for appreciating our study and the constructive remarks. We would like to address the comments point by point. Page and line numbers refer to the revised version of the manuscript.

Reviewer’s comment 1: Sample size calculation:

How many patients were assumed per GP? 

Authors’ reply 1: We appreciate that the reviewer raises several points about our power analysis, which we did not perform according to the perhaps more traditional paradigm of considering the (minimum) sample size as a function of various assumptions about the study (in particular, effect sizes and a specific statistical power to be achieved), but rather considered the achievable power given a potential maximum recruitable sample size. For simple research designs, such a power analysis can be achieved by choosing the appropriate sample size formula that relates the different quantities involved and solving the corresponding equation for the statistical power as a function of the other variables. However, the complex design of our study, which includes a pre- and a post-intervention period, repeated measurements, and potentially highly variable numbers of patients per GP, means that, to the best of our knowledge, no reliable formula is available for our specific situa-tion. This is the main reason why we resorted to a simulation-based power analysis.

Formulas for sample size calculation in randomized trials with hierarchical data structures (often in the context of cluster-randomized trials) typically include inflation factors derived from summaries of the group sizes involved [1], which is where we guess the reviewer’s question originates. Our simulation-based approach circumvents the need to make such assumptions by directly incorporat-ing the expected distribution of patient group sizes per GP from the available dataset into the sam-pling procedure (specifically, the bootstrap sampling step # 1 at the beginning of each iteration also includes the denominator population size, see S6 Appendix). We have clarified this aspect in the corresponding section of the manuscript as follows:

Page 13, lines 306-308: Relevant GP characteristics, including the adjustment covariates planned for the final analysis and the size of the denominator populations, were accounted for by boot-strapping from the observed sample of 64 GPs.

Reviewer’s comment 2: 2% difference is small relative to 50 GPs. 2% difference needs to be clinically justified.

Authors’ reply 2: We agree that, without further consideration, a sample size of 50 GPs may seem small to achieve sufficient statistical power to detect an absolute outcome change of 2 percentage points. However, we note that the planned data analysis and the power calculation took into ac-count the presence of measurements at the level of individual care events, meaning that one could consider the total denominator population size across all recruited GPs as the effective sample size of measurement units. As observed in the sample of 64 GPs used for the power calculation, this effective sample size is easily in the order of magnitude of several tens of thousands of patients.

We agree that the assumption an absolute change in quality indicator performance of 2 percentage points seems arbitrary. Indeed, recent studies examining the effect of audit and feedback interven-tions on the performance on quality indicators similar to those examined in our trial have often arbitrarily assumed relevant performance differences of 10 percentage points in their power calcu-lations [2-5]. Our assumption of a 2 percentage point increase is much lower and thus provides a much more conservative power estimate, demonstrating that our study achieves satisfactory statisti-cal power even for a potentially very small effect size in the primary outcome. We have included this aspect in the manuscript:

Page 13, lines 302-306: For the differential effect of the QI-FBR versus the P-FBR, we assumed an absolute increase of 2 percentage points in the overall QI performance (corresponding to an OR of 1.09 in the observed sample), which is a small effect size compared to the 10 percentage point im-provement often assumed in power analyses of previous studies involving performance feedback on similar QIs [2-5].

Reviewer’s comment 3: What assumptions were used for random factors?

Authors’ reply 3: We presume that ‘random factors’ refers to the random effect terms of the mixed-effects models. As usual in mixed-effects models, these random intercepts are assumed to be iid samples from normal distributions of unknown variance parameters [6]. While this assump-tion is evident in the description of the power analysis of S6 Appendix, it is not mentioned in the main text. We therefore changed the corresponding section of the manuscript as follows:

Page 12, lines 276-278: We will use a mixed-effects model to account for each GP contributing two observations to the dataset (normally distributed GP-level random intercepts) and to account for unobserved effects on the practice level (normally distributed practice-level random intercepts) [6].

Reviewer’s comment 4: Analysis:

What mixed model will you use? GEE or generalized linear mixed model?

Authors’ reply 4: As described in the ‘Statistical analysis’ section of the protocol, the main analy-sis will be based on a binomial mixed-effects regression model. By its common definition, a bino-mial mixed-effects model is simply a generalized linear mixed-effects model with a logistic link function used for discrete outcome variables that can be viewed, for each observation, as the num-bers of successes and failures (or, equivalently, of successes and total trials) of a specific binary outcome. For readers not familiar with the specifics of mixed-effects models, we included a refer-ence providing more background in the appropriate section of the manuscript:

Page 12, lines 276-278: We will use a mixed-effects model to account for each GP contributing two observations to the dataset (normally distributed GP-level random intercepts) and to account for unobserved effects on the practice level (normally distributed practice-level random intercepts) [6].

We also note that in the common terminology, generalized estimating equations (GEEs) do not de-note a class of mixed-effects model, but rather refer to a class of estimation approaches for gener-alized linear models that aim to preserve the unbiasedness and consistency of regression coefficient estimates even under misspecification of unobserved correlation arising from a hierarchical struc-ture of the data [7]. The interpretations of coefficient estimates derived using a mixed-effects ap-proach and a GEE approach differ in that the former correspond to individual-level effects, while the latter indicate population-level effects [7]. In our specific example, we opted for an interpreta-tion of the various effects linked to the presence of feedback reports and to the inclusion of quality indicator performance as individual-level effects, and thus preferred mixed-effects modeling over a GEE approach.

Reviewer’s comment 5: Is there any multiple testing correction for secondary endpoints?

Authors’ reply 5: We recognize that the regression model planned for the secondary analysis will involve an individual hypothesis tests for each of the quality indicators included in the feedback reports. The question of addressing whether the study intervention affects performance of any of the quality indicators would be prone to increased family-wise error rate when considering the indi-vidual associated p-values without appropriate adjustment. We would therefore include a specific global hypothesis test as a comparison of the model that does and the model that does not include each quality indicator. In other words: We will consider the p-value associated with the inclusion of the quality indicator-specific interaction term. A similar approach will be used for the different do-mains of care. We have updated the corresponding section of the manuscript accordingly:

Page 14, Lines 322-324: We will address the global hypotheses of whether the QI-FBR is effective on performance in any of the chronic condition domains or in any of the QIs by considering the p-values associated with the inclusion of the respective interaction terms.

Reviewer’s comment 6: Missing:

Can you increase the sample size to allow for dropouts instead of imputation? (e.g., if expecting 10% dropout, recruit 55 GPs instead of 50)

Authors’ reply 6: We are happy to include consideration of dropouts in the sample size calcula-tion. In the previous study on the quality of diabetes care conducted within the FIRE project and mentioned in the protocol, 6 out of 71 GPs dropped out during a 12-month period [8]. A dropout rate of 10% therefore seems a reasonable estimate, resulting in a target sample size of 56 GPs. We have updated the manuscript accordingly:

Page 14, Lines 311-314: Assuming a drop-out rate of 10%, which is slightly higher than that ob-served in the aforementioned previous trial within the FIRE project [8], a sample size of 56 GPs would ensure a statistical power of 88% to detect a 2 percentage point increase in overall QI per-formance across all GPs.

We also note that aiming for a larger sample size that accounts for potential dropouts does not re-place the usefulness of imputation methods, which we intend to use in an additional analysis to address potential biases introduced by missing data and to compensate for a potential reduction in statistical power due to a reduced sample size. We therefore decided to retain this additional impu-tation analysis, especially considering that we aim to ensure transparency by presenting both the results without and with imputation.

References

1. Rutterford C, Copas A, Eldridge S. Methods for sample size determination in cluster randomized trials. Int J Epidemiol. 2015;44(3):1051-67. Epub 20150713. doi: 10.1093/ije/dyv113. PubMed PMID: 26174515; PubMed Central PMCID: PMC4521133.

2. Patel MS, Kurtzman GW, Kannan S, Small DS, Morris A, Honeywell S, Jr., et al. Effect of an Automated Patient Dashboard Using Active Choice and Peer Comparison Performance Feedback to Physicians on Statin Prescribing: The PRESCRIBE Cluster Randomized Clinical Trial. JAMA Netw Open. 2018;1(3):e180818. Epub 20180706. doi: 10.1001/jamanetworkopen.2018.0818. PubMed PMID: 30646039; PubMed Central PMCID: PMC6324300.

3. Peiris D, Usherwood T, Panaretto K, Harris M, Hunt J, Redfern J, et al. Effect of a Computer-Guided, Quality Improvement Program for Cardiovascular Disease Risk Management in Primary Health Care. Circulation: Cardiovascular Quality and Outcomes. 2015;8(1):87-95. doi: 10.1161/CIRCOUTCOMES.114.001235. PubMed PMID: 25587090.

4. Meier R, Muheim L, Senn O, Rosemann T, Chmiel C. The impact of financial incentives to improve quality indicators in patients with diabetes in Swiss primary care: a protocol for a cluster randomised controlled trial. BMJ Open. 2018;8(6):e023788. doi: 10.1136/bmjopen-2018-023788. PubMed PMID: 29961043; PubMed Central PMCID: PMC6042619.

5. Vinereanu D, Lopes RD, Bahit MC, Xavier D, Jiang J, Al-Khalidi HR, et al. A multifaceted intervention to improve treatment with oral anticoagulants in atrial fibrillation (IMPACT-AF): an international, cluster-randomised trial. Lancet. 2017;390(10104):1737-46. doi: 10.1016/s0140-6736(17)32165-7. PubMed PMID: 28859942.

6. Pinheiro JC, Bates DM. Mixed-Effects Models in S and S-PLUS. New York: Springer; 2000.

7. Gardiner JC, Luo Z, Roman LA. Fixed effects, random effects and GEE: what are the differences? Stat Med. 2009;28(2):221-39. doi: 10.1002/sim.3478. PubMed PMID: 19012297.

8. Meier R, Chmiel C, Valeri F, Muheim L, Senn O, Rosemann T. The Effect of Financial Incentives on Quality Measures in the Treatment of Diabetes Mellitus: a Randomized Controlled Trial. J Gen Intern Med. 2022;37(3):556-64. Epub 20210426. doi: 10.1007/s11606-021-06714-8. PubMed PMID: 33904045; PubMed Central PMCID: PMC8858366.

---

## [Decision Letter · Decision Letter 1]

11 Nov 2024

The Effectiveness of a multi-domain electronic feedback report on the performance of quality indicators for chronic conditions: protocol for a randomized controlled trial in general practice

PONE-D-24-36573R1

Dear Dr. Jäger,

We’re pleased to inform you that your manuscript has been judged scientifically suitable for publication and will be formally accepted for publication once it meets all outstanding technical requirements.

Kind regards,

Kathleen Finlayson

Academic Editor

PLOS ONE

Additional Editor Comments (optional):

Reviewers' comments:

Reviewer's Responses to Questions

**Comments to the Author**

1. Does the manuscript provide a valid rationale for the proposed study, with clearly identified and justified research questions?

Reviewer #2: Yes

2. Is the protocol technically sound and planned in a manner that will lead to a meaningful outcome and allow testing the stated hypotheses?

Reviewer #2: Yes

3. Is the methodology feasible and described in sufficient detail to allow the work to be replicable?

Reviewer #2: Yes

4. Have the authors described where all data underlying the findings will be made available when the study is complete?

Reviewer #2: Yes

5. Is the manuscript presented in an intelligible fashion and written in standard English?

Reviewer #2: Yes

6. Review Comments to the Author

You may also provide optional suggestions and comments to authors that they might find helpful in planning their study.

Reviewer #2: All my concerns were addressed.

7. PLOS authors have the option to publish the peer review history of their article (what does this mean?). If published, this will include your full peer review and any attached files.

Reviewer #2: No

---

## [Editor Report · Acceptance letter]

13 Nov 2024

PONE-D-24-36573R1 

PLOS ONE

Dear Dr. Jäger, 

I'm pleased to inform you that your manuscript has been deemed suitable for publication in PLOS ONE. Congratulations! Your manuscript is now being handed over to our production team.

Kind regards, 

on behalf of

Dr. Kathleen Finlayson 

Academic Editor

PLOS ONE